# Optimizing Antimicrobial Peptide Design: Integration of Cell-Penetrating Peptides, Amyloidogenic Fragments, and Amino Acid Residue Modifications

**DOI:** 10.3390/ijms25116030

**Published:** 2024-05-30

**Authors:** Sergey V. Kravchenko, Pavel A. Domnin, Sergei Y. Grishin, Alena P. Zakhareva, Anastasiia A. Zakharova, Leila G. Mustaeva, Elena Y. Gorbunova, Margarita I. Kobyakova, Alexey K. Surin, Darya V. Poshvina, Roman S. Fadeev, Viacheslav N. Azev, Olga S. Ostroumova, Svetlana A. Ermolaeva, Oxana V. Galzitskaya

**Affiliations:** 1Institute of Environmental and Agricultural Biology (X-BIO), Tyumen State University, 625003 Tyumen, Russia; svkraft@yandex.ru (S.V.K.); heggiesan@gmail.com (A.P.Z.); dvposhvina@mail.ru (D.V.P.); 2Biology Faculty, Lomonosov Moscow State University, 119991 Moscow, Russia; paveldomnin6@gmail.com; 3Gamaleya Research Centre of Epidemiology and Microbiology, 123098 Moscow, Russia; drermolaeva@mail.ru; 4Institute of Protein Research, Russian Academy of Sciences, 142290 Pushchino, Russia; syugrishin@gmail.com (S.Y.G.); alan@vega.protres.ru (A.K.S.); 5Institute of Cytology of the Russian Academy of Sciences, Tikhoretsky Ave. 4, 194064 St. Petersburg, Russia; zaza2187@bk.ru (A.A.Z.); osostroumova@mail.ru (O.S.O.); 6The Branch of the Institute of Bioorganic Chemistry, Russian Academy of Sciences, 142290 Pushchino, Russia; mustaeva@rambler.ru (L.G.M.); eyugorbunova@rambler.ru (E.Y.G.); viatcheslav.azev@bibch.ru (V.N.A.); 7Institute of Theoretical and Experimental Biophysics, Russian Academy of Sciences, 142290 Pushchino, Russia; kobyakovami@gmail.com (M.I.K.); fadeevrs@gmail.com (R.S.F.); 8Research Institute of Clinical and Experimental Lymphology—Branch of the Institute of Cytology and Genetics, Siberian Branch of Russian Academy of Sciences, 630060 Novosibirsk, Russia; 9State Research Center for Applied Microbiology and Biotechnology, 142279 Obolensk, Russia

**Keywords:** antimicrobial peptides, amyloidogenic regions, cell-penetrating peptide, TAT fragment, Antp fragment, ribosomal S1 protein

## Abstract

The escalating threat of multidrug-resistant pathogens necessitates innovative approaches to combat infectious diseases. In this study, we examined peptides R23F^S^*, V31K^S^*, and R44K^S^*, which were engineered to include an amyloidogenic fragment sourced from the S1 protein of *S. aureus*, along with one or two cell-penetrating peptide (CPP) components. We assessed the antimicrobial efficacy of these peptides in a liquid medium against various strains of both Gram-positive bacteria, including *S. aureus* (209P and 129B strains), MRSA (SA 180 and ATCC 43300 strains), and *B. cereus* (strain IP 5832), and Gram-negative bacteria such as *P. aeruginosa* (ATCC 28753 and 2943 strains) and *E. coli* (MG1655 and K12 strains). Peptides R23F^S^*, V31K^S^*, and R44K^S^* exhibited antimicrobial activity comparable to gentamicin and meropenem against all tested bacteria at concentrations ranging from 24 to 48 μM. The peptides showed a stronger antimicrobial effect against *B. cereus*. Notably, peptide R44K^S^* displayed high efficacy compared to peptides R23F^S^* and V31K^S^*, particularly evident at lower concentrations, resulting in significant inhibition of bacterial growth. Furthermore, modified peptides V31K^S^* and R44K^S^* demonstrated enhanced inhibitory effects on bacterial growth across different strains compared to their unmodified counterparts V31K^S^ and R44K^S^. These results highlight the potential of integrating cell-penetrating peptides, amyloidogenic fragments, and amino acid residue modifications to advance the innovation in the field of antimicrobial peptides, thereby increasing their effectiveness against a broad spectrum of pathogens.

## 1. Introduction

The treatment of diseases caused by multidrug-resistant bacteria remains a significant concern, demanding a comprehensive approach that integrates the use of well-established antibiotics alongside the creation of novel antimicrobial agents [1,2]. Interest in diverse therapeutic agents, including antimicrobial peptides (AMPs), for infectious diseases is growing, with a particular focus on combating multidrug-resistant pathogens such as *S. aureus* and *P. aeruginosa* [3,4]. AMPs are peptide molecules that play a significant role in the innate immune response of various organisms [5,6]. In most cases, the use of antimicrobial peptides is considered in pharmaceuticals; however, some AMPs can be developed for use in the cosmeceutical industry [7], food industry [8], agriculture [9], and even in the textile industry [10].

AMPs can bring significant benefits to public health by combating resistant pathogens and alleviating incurable diseases; however, their drawbacks, such as enzymatic degradation and challenges in designing them for specific targets, pose serious obstacles [11]. Exploring the compatibility of synthetic AMPs with cell structures elucidates their suitability and adaptability to host cells. In this context, creating artificial AMPs may offer certain advantages compared to natural AMPs. Pathogenic bacteria can adapt to natural AMPs by producing corresponding proteases that cleave the peptide sequence, thereby inactivating the peptide and preventing it from exerting its antimicrobial action. Synthetic AMPs containing modified amino acid residues may be more resistant to cellular proteases and thus more stable compared to natural AMPs [12]. Modifications of the peptide’s amino acid composition through non-canonical residues can reduce the availability of binding sites for bacterial proteases, making the modified AMP more resistant to proteolytic hydrolysis and thus increasing its stability and antimicrobial activity. Therefore, pathogenic bacteria are less protected against such modified synthetic AMPs compared to unmodified AMPs. Synthetic AMPs can be engineered to possess multiple antimicrobial mechanisms, thereby impeding the emergence of drug-resistant bacteria [13,14]. On the other hand, rational design allows for the creation of AMP sequences that are directed against specific bacterial proteins [15,16,17]. One of the promising approaches is the utilization of relatively short amino acid sequences from proteins as antimicrobial peptides [18]. These short peptide sequences may exhibit better characteristics in terms of charge, structure, and overall antimicrobial effect compared to whole antimicrobial proteins [19]. In turn, some peptides designed in silico, modified, and synthesized may demonstrate antimicrobial effects comparable to or even better than those of natural counterparts [20]. Designing new synthetic peptides can be used not only to enhance their antimicrobial effects but also for broader applications in various fields [21]. With the proper design and selection of amino acid sequences, synthetic peptides can be integrated into the surface matrix or the material of the product itself, thereby providing long-lasting protection against microbes [22,23,24]. 

Conjugating AMPs with cell-penetrating peptides (CPPs) [25] or antibiotics [26] enhances their antibacterial activity, with L-glutamine in particular demonstrating the ability to raise the efficacy of relevant antimicrobials [27], arising from its fundamental structure and properties [28]. CPPs demonstrate potential in drug and vaccine delivery systems, especially when conjugated with peptides [29]. Moreover, they possess advantages such as ease of synthesis and stability, low toxicity, and high specificity, making them promising candidates for therapeutic interventions [30,31]. Adding CPP to the peptide facilitates the rapid penetration of the AMP into the cell and may enhance its effectiveness against pathogenic microorganisms [32]. Due to their peptide nature, synthetic AMPs with CPPs can be biocompatible with direct drug delivery systems and drug-releasing devices [33].

Recently, several strategies have been proposed involving the targeting of peptides to amyloidogenic proteins or specific amyloidogenic regions of proteins [34,35]. Studying the mechanisms of amyloid structure formation in antimicrobial peptides can provide a foundation for developing effective peptide-based antimicrobial agents [36]. Recent studies demonstrate the potential importance of amyloidogenic AMPs in innate immune responses against microbial infections [37,38]. Some self-aggregating antimicrobial peptides possess antibiofilm activity [39]. The antimicrobial properties and propensity for self-aggregation of antimicrobial peptides also depend significantly on their amino acid composition [40].

Typically, modification of amino acid residues in AMPs or the creation of hybrid constructs based on AMPs can lead to changes in their minimum inhibitory concentration as well as their aggregation properties [41,42]. Incorporating non-natural amino acids into the composition of AMPs may improve proteolytic stability and cytocompatibility with human cells [43,44]. However, this is not always the case, as there are known instances where the high effectiveness of an antimicrobial peptide against bacteria was associated with cytotoxicity towards human cells [45,46]. Overall, the incorporation of non-canonical amino acids into the composition of AMPs is currently a common approach to diversify their physicochemical characteristics for subsequent screening in preclinical studies against pathogenic microorganisms [47].

In this study, we aimed to evaluate the antimicrobial efficacy of peptides engineered using fragments of cell-penetrating peptides and amyloidogenic amino acid segments from the ribosomal protein S1 of *S. aureus*, and incorporated amino acid modifications. As a basis for the amyloidogenic region of the peptides, the amyloidogenic fragment VVVHINGGKF from the S1 protein of *S. aureus* was chosen [48]. In addition to the amyloidogenic region, peptides R23F^S^*, V31K^S^, V31K^S^*, R44K^S^, and R44K^S^* contained one (TAT fragment or Antp fragment [49]) or two (TAT fragment and Antp fragment) cell-penetrating peptides—RKKRRQRRR and SRQIKIWFQNRRMKWKK, respectively. Finally, the peptides underwent amino acid modifications, including amidation of the C-terminus of the peptide chain and the replacement of canonical amino acid residues with other canonical or non-canonical amino acid residues (X). We evaluated the antimicrobial effectiveness of these peptides in a liquid medium against various strains of both Gram-positive bacteria, including *S. aureus* (209P and 129B strains), MRSA (SA 180 and ATCC 43300 strains), and *B. cereus* (strain IP 5832), and Gram-negative bacteria such as *P. aeruginosa* (ATCC 28753 and 2943 strains) and *E. coli* (MG1655 and K12 strains). Additionally, we tested the cytotoxic properties of peptides against human fibroblasts and lung carcinoma cells, as well as their effect on the ion permeability of lipid bilayers.

## 2. Results

### 2.1. Antibacterial Activity of Peptides V31K^S^, R44K^S^, V31K^S^*, and R44K^S^* against Gram-Positive and Gram-Negative Bacteria in Liquid Medium 

We tested unmodified peptides, V31K^S^ and R44K^S^, as well as peptides with modifications, V31K^S^* and R44K^S^*, which were designed to contain an amyloidogenic fragment from the S1 protein of *S. aureus*, as well as one or two cell-penetrating peptide (CPP) fragments. We evaluated the antimicrobial effects of peptides V31K^S^, R44K^S^, V31K^S^*, and R44K^S^* in a liquid medium against Gram-positive bacteria *S. aureus* (209P and 129B strains) and MRSA (SA 180 and ATCC 43300 strains), as well as Gram-negative bacteria *P. aeruginosa* (ATCC 28753 and 2943 strains) and *E. coli* (MG1655 and K12 strains).

Initially, we assessed the antimicrobial effects of peptides V31K^S^, R44K^S^, V31K^S^*, and R44K^S^* in a liquid medium against Gram-positive *S. aureus* (209P and 129B strains) (Figure 1).

As shown in Figure 1, modified peptides V31K^S^* and R44K^S^* generally exhibited a more pronounced inhibitory effect on the growth of bacteria from different strains compared to the unmodified V31K^S^ and R44K^S^. For the unmodified peptide V31K^S^, complete inhibition of *S. aureus* strains was not demonstrated. However, for peptide V31K^S^*, inhibition of bacterial growth comparable to the effect of gentamicin was observed at a concentration of 12 µM. An inhibitory effect was also observed for concentrations of 1.5–12 µM of the peptide R44K^S^* against *S. aureus* strain 209P. Conversely, the unmodified peptide R44K^S^ demonstrated antimicrobial effects against the same strain at concentrations of 6–12 µM. Thus, the antimicrobial effect of the modified peptide was evident at a lower concentration.

We checked the antimicrobial effects of peptides V31K^S^, R44K^S^, V31K^S^*, and R44K^S^* in a liquid medium with Gram-positive MRSA (ATCC 43300 and SA 180 strains) (Figure 2).

As shown in Figure 2, in general, the inhibitory effect on bacterial growth from various strains was more pronounced with the modified peptides V31K^S^* and R44K^S^* than with the unmodified versions V31K^S^ and R44K^S^. An inhibitory effect was observed for concentrations of 6–12 µM of peptide R44K^S^*, indicating a high antimicrobial effect of the peptide against MRSA, strain ATCC 43300. Meanwhile, for the unmodified peptide R44K^S^, the concentration at which bacterial growth was completely suppressed for 15 h was 12 µM.

We assessed the antimicrobial properties of peptides V31K^S^, R44K^S^, V31K^S^*, and R44K^S^* in a liquid medium containing Gram-negative *P. aeruginosa* (ATCC 28753 and 2943 strains) (Figure 3).

As shown in Figure 3, during the incubation of peptides V31K^S^, R44K^S^, V31K^S^*, and R44K^S^*, the peptides exhibited a stronger antimicrobial effect against the *P. aeruginosa* ATCC 28753 strain compared to the 2943 strain. The concentrations of peptides suppressing the growth of bacteria of the ATCC 28753 strain, compared to the negative control, were 6 μM, 6 μM, 1.5 μM, and 1.5 μM for peptides V31K^S^, V31K^S^*, R44K^S^, and R44K^S^*, respectively. Meanwhile, peptides V31K^S^ and R44K^S^ did not exhibit antimicrobial effects against the *P. aeruginosa* 2943 strain compared to the negative control. Thus, the inhibitory effect on the growth of bacteria from different strains was more pronounced with the modified peptides V31K^S^* and R44K^S^* than with the unmodified versions V31K^S^ and R44K^S^.

We examined the antimicrobial properties of peptides V31K^S^, R44K^S^, V31K^S^*, and R44K^S^* in a liquid medium containing Gram-negative *E. coli* strains (K12 and MG1655) (Figure 4).

Figure 4 illustrates that over the incubation period, peptide V31K^S^ did not exhibit antimicrobial activity against *E. coli* (K12 strain), and only a weak antimicrobial effect was observed against the MG1655 strain at a concentration of 276.8 μM. Conversely, the modified peptide V31K^S^* displayed antimicrobial activity against the same strains, K12 and MG1655, at concentrations of 3 μM and 276.8 μM, respectively. Additionally, amino acid modifications enhanced the antimicrobial effect of peptide R44K^S^* against the K12 and MG1655 strains, resulting in a decrease in effective concentrations from 6 to 0.75 μM and from 19.4 to 1.9 μM compared to the unmodified peptide R44K^S^.

### 2.2. Antibacterial Efficacy of Peptides R23F^S^*, V31K^S^*, and R44K^S^* against Both Gram-Positive and Gram-Negative Bacteria in a Liquid Medium

We tested peptides R23F^S^*, V31K^S^*, and R44K^S^*, which were designed to contain an amyloidogenic fragment from the S1 protein of *S. aureus*, as well as one or two cell-penetrating peptide (CPP) fragments. We evaluated the antimicrobial effects of peptides R23F^S^*, V31K^S^*, and R44K^S^* in a liquid medium against Gram-positive bacteria *S. aureus* (209P and 129B strains), MRSA (SA 180 and ATCC 43300 strains), and *B. cereus* (strain IP 5832), as well as Gram-negative bacteria *P. aeruginosa* (ATCC 28753 and 2943 strains) and *E. coli* (MG1655 and K12 strains) (Figure 5).

As shown in Figure 5, peptides R23F^S^*, V31K^S^*, and R44K^S^* at concentrations of 24–48 μM exhibited an antimicrobial effect comparable to gentamicin (100 μM) against all bacteria. The strongest antimicrobial effect of the peptides was observed against *B. cereus*. Peptide R44K^S^* was more effective compared to peptides R23F^S^* and V31K^S^*, as observed at lower concentrations inhibiting bacterial growth.

As depicted in Figure 6, the most effective peptide against *S. aureus* (strain 129B), MRSA (strain SA 180), *P. aeruginosa* (strain 2943), and *E. coli* (strain MG1655) was peptide R44K^S^*.

### 2.3. Antibacterial Effects of Peptides against S. aureus, MRSA, P. aeruginosa, and E. coli

We assessed the peptides’ antimicrobial efficacy against various strains of *S. aureus*, MRSA, *P. aeruginosa*, and *E. coli* bacteria. Apart from the previously mentioned data concerning susceptible strains 209P (*S. aureus*), ATCC 43300 (MRSA), strain IP 5832 (*B. cereus*), ATCC 28753 (*P. aeruginosa*), and K12 (*E. coli*), we also evaluated their effects on clinical isolate strains 129B (*S. aureus*), SA 180-F (MRSA), and 2943 (*P. aeruginosa*), as well as susceptible MG1655 (*E. coli*). Table 1 presents comparative results for two strains of each organism.

According to Table 1 and as a result of comparing the effect of peptides on two different strains of each of the microorganisms (*S. aureus*, MRSA, *B. cereus*, *E. coli*, *P. aeruginosa*), it can be concluded that the highest antimicrobial effect was shown by peptide R44K^S^*.

### 2.4. Effect of R44K^S^ on Ionic Permeability of Lipid Bilayers

The effects of peptide R44K^S^ on the ionic permeability of planar lipid bilayers composed of an equimolar mixture of POPE and POPG mimicking target bacterial cell membranes was examined. The example of the time dependence of the transmembrane current produced by a subsequent one-side addition of R44K^S^ into the bathing solution (0.1 M KCl, pH 7.4) is shown in Figure 7A. 

An increase in the peptide concentration up to 40 μg/mL affected membrane stability and triggered its rupture, causing a detergent action. As a rule, at subthreshold concentrations, R44K^S^ was able to induce single step-like current fluctuations (inset in Figure 7A), which should be attributed to ion-permeable transmembrane pores. Figure 7B shows that the current fluctuations related to the functioning of R44K^S^-produced pores were observed at various transmembrane voltages. The single pores were characterized by different amplitudes with conductance varying in the range of 10–30 pS. The variability in conductance states could be attributed to the formation of barrel oligomers by R44K^S^, as predicted by AlphaFold 2 (Figure 8).

### 2.5. Toxicity of R23F^S^*, V31K^S^, R44K^S^, V31K^S^*, and R44K^S^* against Eukaryotic Cells

To analyze the cytotoxic effect, cells after incubation with peptides were stained with calcein AM (stains living cells) and propidium iodide (stains dead cells). It was shown that the peptides V31K^S^, R44K^S^, and V31K^S^*, at a concentration of 100 µg/mL (both with and without pre-incubation), had no cytotoxic effect on BT474 cells and human skin fibroblasts after 96 h of incubation (Figure 9 and Figure 10). At the same time, the peptide R23F^S^* in the concentration range of 100–50 µg/mL (both with and without pre-incubation) had a weak cytotoxic effect on human skin fibroblasts after 96 h of incubation (Figure 8). The number of dead cells was 12 ± 3% and 11 ± 3%, respectively. The number of dead BT474 cells, at concentrations of 100, 50, and 25 µg/mL (both with and without pre-incubation), was 20 ± 5%, 16 ± 4%, and 12 ± 3%, respectively (Figure 10). Peptide R44K^S^* at a concentration of 100 µg/mL (without pre-incubation) had a strong cytotoxic effect on BT474 cells (95 ± 4% of dead cells) and human skin fibroblasts (84 ± 7% of dead cells) after 96 h of incubation. But at an R44K^S^* peptide concentration of 50 µg/mL (without pre-incubation), the number of dead BT474 cells and human skin fibroblasts was 22 ± 5% and 15 ± 5%, respectively. However, the same peptide with pre-incubation had no cytotoxic effect on BT474 cells and human skin fibroblasts after 96 h of incubation (Figure 9 and Figure 10).

Analysis of the cytostatic effect was performed on the basis of the distribution of cells by phases of the cell cycle and mitotic activity. It was shown that all the studied peptides, both with and without pre-incubation, did not suppress the proliferative activity of viable BT-474 cells and human skin fibroblasts.

## 3. Discussions

The importance of developing new synthetic antimicrobial peptides (AMPs) to combat bacterial resistance to traditional antibiotics is driven by their advantages over natural peptides, such as stability, specificity, and multifunctionality [51,52,53,54,55]. During the development of new synthetic AMPs, researchers often find inspiration in the structure or mechanisms of action of natural antimicrobial peptides. In particular, several strategies have recently been proposed that leverage the amyloidogenic properties of known antimicrobial peptides to create novel AMPs capable of targeting the membranes or intracellular protein targets of pathogenic bacteria [15,56,57]. On the other hand, the potential of cell-penetrating peptides (CPPs) to deliver cargo into cells has been demonstrated, along with the prospects of combining CPPs with other AMPs to enhance the antimicrobial properties of such hybrid peptides [14,58,59]. Lastly, amino acid modifications of antimicrobial peptides have also allowed for the improvement of their stability, specificity, and activity against microorganisms [13,60]. They can also reduce the likelihood of development of bacterial resistance to AMPs [61].

In this study, we evaluated the antimicrobial potential of peptides derived from the amyloidogenic fragment VVVHINGGKF of the S1 protein from *S. aureus*. Penetrating peptides, including the TAT fragment and the Antp fragment, were utilized both individually (TAT fragment for the design of R23F^S^* and Antp fragment for the design of V31K^S^ and V31K^S^*) and in combination (TAT fragment at the N-terminus, Antp at the C-terminus for the design of R44K^S^ and R44K^S^*) to create novel peptides. Modifications included amide formation at the C-terminus and the substitution of canonical amino acids with non-canonical ones. Amino acid modifications, especially in peptides V31K^S^* and R44K^S^*, led to enhanced suppression of bacterial growth across multiple strains. Moreover, peptides containing the TAT fragment exhibited greater efficacy in inhibiting bacterial growth compared to those containing only the Antp penetration peptide fragment. This was demonstrated in our recent article, where we tested peptides based on the amyloidogenic region of the S1 ribosomal protein of *P. aeruginosa* [62]. Notably, peptide R44K^S^* displayed the most potent antimicrobial activity against a wide range of microorganisms, but also exhibited higher toxicity towards human cells. The peptides R23F^S^*, V31K^S^, R44K^S^, V31K^S^*, and R44K^S^* demonstrated a strain-specific antimicrobial effect against various bacteria, including *S. aureus*, MRSA, *P. aeruginosa*, and *E. coli*. These peptides showed a superior ability to inhibit the growth of bacterial strains compared to conventional antibiotics, particularly at lower concentrations. Overall, our study demonstrates the effectiveness of combining amyloidogenic regions from the S1 protein with penetration peptide fragments and amino acid modifications to develop peptides with improved antimicrobial properties.

In our study, we evaluated the antimicrobial activity of peptides V31K^S^, R44K^S^, V31K^S^*, and R44K^S^* against various bacterial strains. The antimicrobial effects of peptides appear to be unrelated to Gram-positive or Gram-negative classification, but rather strongly depend on the bacterial strain of the same species. Modified peptides V31K^S^* and R44K^S^* demonstrated enhanced inhibitory effects compared to their unmodified counterparts. Specifically, peptide V31K^S^* exhibited inhibition comparable to gentamicin against *S. aureus* strains at a concentration of 12 µM, while peptide R44K^S^* showed inhibitory effects against *S. aureus* strain 209P at concentrations as low as 3–12 µM, compared to 6–12 µM for unmodified peptide R44K^S^. Our investigation into the antimicrobial effects of peptides against Gram-positive MRSA strains revealed that modified peptides exhibited a more pronounced inhibitory effect on bacterial growth compared to their unmodified counterparts. Specifically, peptide R44K^S^* demonstrated notable antimicrobial activity against MRSA strain ATCC 43300 at concentrations of 6–12 µM, indicating its potential efficacy. When evaluating peptides against Gram-negative *P. aeruginosa* strains, differences in effectiveness were observed, with modified peptides showing enhanced efficacy against the ATCC 28753 strain compared to the 2943 strain. Furthermore, our examination of peptides against Gram-negative *E. coli* strains revealed varying degrees of antimicrobial activity, with modified peptides demonstrating improved efficacy at lower concentrations. These findings suggest the potential of modified peptides in combating bacterial infections, albeit with variations in effectiveness against different bacterial strains. We assessed peptides R23F^S^*, V31K^S^*, and R44K^S^* for their antimicrobial effects against various bacteria strains. Peptides R23F^S^*, V31K^S^*, and R44K^S^* at concentrations of 24–48 μM exhibited antimicrobial activity comparable to gentamicin (100 μM) against all bacteria strains, with the strongest effect observed against *B. cereus*. Peptide R44K^S^* demonstrated superior efficacy, inhibiting bacterial growth at lower concentrations compared to R23F^S^* and V31K^S^*.

The mechanism of action of peptide R44K^S^ may be partially or entirely attributed to pore formation in bacterial membranes. This is indicated by the observed data on step-like current fluctuations associated with the formation of ion-permeable transmembrane pores. Previously, Kravchenko et al. showed that the synthetic R23I^T^ peptide containing the TAT sequence (RKKRRQRRR) at the N-terminus of the amyloidogenic fragment caused the disruption of POPE:POPG bilayers at concentrations greater than 80 μg/mL [62]. The obtained difference in the concentrations of R23I^T^ and R44K^S^ peptides required for membrane breakdown might be explained by the additional CPP fragment (the antennapedia peptide fragment (SRQIKIWFQNRRMKWKK)) on the C-terminus of the amyloidogenic peptide of R44K^S^. Probably, the presence of two CPP fragments enhances the membrane activity of R44K^S^ compared to R23I^T^ [62], and moreover R44K^S^, demonstrated greater antibacterial activity than R23I^T^.

Based on the models predicted by AlphaFold 2, it can be suggested that peptide R44K^S^ may have a tendency to form oligomers. In turn, it is often oligomeric forms of amyloidogenic and antimicrobial peptides that are responsible for the formation of pores in bacterial cell membranes [63].

Peptides R23F^S^*, V31K^S^, R44K^S^, V31K^S^*, and R44K^S^* were tested for cytotoxic effects on human fibroblast cells and the BT474 lung carcinoma cell line. It was observed that peptide R23F^S^* exhibited moderate cytotoxicity against both cell lines, while peptide R44K^S^* showed strong cytotoxic effects, particularly without pre-incubation. Peptides V31K^S^, V31K^S^*, and R44K^S^ did not significantly affect cell viability. These findings highlight the varying cytotoxic potentials of the tested peptides, with R44K^S^* demonstrating the most notable effects.

These results suggest that modifications to the peptides enhanced their antimicrobial efficacy at lower concentrations. However, it is important to note that our assessment of the feasibility of the results is limited by the available data. Some aspects, such as the impact of peptides on different bacterial strains and their toxicity to human cells, require further research and confirmation. Additionally, potential limitations, such as peptide synthesis and stability complexities, as well as their possible side effects, need to be considered. In the future, we plan to investigate the synergistic effects of peptides containing various amyloidogenic regions.

## 4. Materials and Methods

### 4.1. Synthesis and Purification of Peptides

The preparation of peptide RKKRRQRRRGG-Sar-GVVVHI-X-GGKF-NH_2_ (R23F^S^*) has been described elsewhere [48]. Peptides VVVHINGGKFGGGGSRQIKIWFQNRRMKWKK (V31K^S^) and RKKRRQRRRGGGGVVVHINGGKFGGGGSRQIKIWFQNRRMKWKK (R44K^S^) were commercial products (IQ Chemical LLC, S. Petersburg, Russia). Boc/Bzl solid-phase synthesis of peptides V31K^S^* (G-VVVHINGGKFGG-Sar-GSRQIKIWFQNRR-X-KWKK-NH_2_) and R44K^S^* (RKK-K-RQRRRGG-Sar-GVVVHINGGKFGG-Sar-GSRQIKIWFQNRR-X-KWKK-NH_2_) was performed using a standard set of protected amino acid derivatives starting with MBHA resin [64]. Protected amino acids were activated with TBTU [65]. The completeness of the coupling reactions was monitored using Kaiser’s ninhydrin test [66]. Whenever aggregation occurred, acylation reactions were repeated using either HFIP-DCM [67] or THF-NMP [68] solvent systems.

Reactions for the global deprotection and cleavage of the peptides from the solid supports were performed using 1M TFMSA/thioanisole in TFA at 20 ºC for 2 h [69]. Crude peptide mixtures were precipitated with anhydrous ether and dried in vacuo over KOH pellets for 18 h. The solids obtained were neutralized with 0.1M aq. NH_4_HCO_3_ and the target peptides were isolated using gel filtration (Sephadex G-10) followed by semi-preparative HPLC in isocratic mode (mobile phase “A”: 0.1% TFA in water; mobile phase “B”: acetonitrile (no additives)) on a Luna C18 250 × 21.5 mm (10 μm) column (Phenomenex, Torrance, CA, USA) at a flow rate of 10 mL/min. The collected fractions were analyzed using RP-HPLC on a Luna 5u C18 (2) 100 Å 250 × 4.6 column (Phenomenex, Torrance, CA, USA). The appropriate fractions were lyophilized and the peptide identity was confirmed using an Orbitrap Elite mass spectrometer (Thermo Scientific, Dreieich, Germany). The observed peptide molecular weight coincided with the calculated one (see Appendix A).

### 4.2. Microorganism Strains

We utilized various strains of microorganisms in this study. For *Staphylococcus aureus*, we employed two strains: a susceptible strain, 209P, and a clinical isolate strain, 129B, which exhibits resistance to multiple antibiotics including clindamycin, oxacillin, erythromycin, sulfamethoxazole, and vancomycin. Additionally, two methicillin-resistant *Staphylococcus aureus* (MRSA) strains were included: a susceptible strain, ATCC 43300, which is resistant to methicillin, oxacillin, and ampicillin, and a clinical isolate strain, SA 180, which demonstrates resistance to ciprofloxacin, clindamycin, erythromycin, oxacillin, sulfamethoxazole, levomycetin, and vancomycin. *Bacillus cereus* strain IP 5832 (susceptible), *Pseudomonas aeruginosa* strains ATCC 28753 (susceptible) and 2943 (resistant), and *Escherichia coli* strains K12 and MG1655 (both susceptible) were also utilized in our experiments.

### 4.3. Antibacterial Activity Assessment of Peptides against S. aureus, MRSA, B. cereus, P. aeruginosa, and E. coli in Liquid Medium

The preparation and co-incubation of the peptides R23F^S^*, V31K^S^, R44K^S^, V31K^S^*, and R44K^S^* with bacteria followed previously established protocols [42]. *S. aureus* (strains 209P and 129B), MRSA (strains ATCC 43300 and SA 180-F), *B. cereus* (strain IP 5832), *P. aeruginosa* (strains ATCC 28753 and 2943), and *E. coli* (strains K12 and MG1655) were cultured on Mueller–Hinton Broth (MHB).

### 4.4. Electrophysiological Technique 

The method of recording the transmembrane current under voltage-clamp conditions was used to study the effect of a synthetic peptide on a lipid membrane. Planar lipid bilayers were formed by the Montal and Mueller technique [70] on an aperture of 50 µm diameter in a 10 µm thick Teflon film separating two compartments of Teflon chambers. Each compartment contained 0.1 M KCl and 10 mM HEPES at pH 7.4. Hexadecane was used for the aperture pretreatment. An equimolar mixture of 1-palmitoyl-2-oleoyl-sn-glycero-3-phosphoethanolamine (POPE) and 1-palmitoyl-2-oleoyl-sn-glycero-3-phospho-(1′-rac-glycerol) (POPG) was chosen to form the lipid bilayers mimicking both Gram-negative and Gram-positive bacteria. The lipids were from Avanti Polar Lipids^®^ (Avanti Polar Lipids, Inc., Ala-baster, AL, USA).

The R44K^S^ peptide was added to the aqueous phase at one side of the membrane at the concentration of 5–40 μg/mL from DMSO stock solutions after bilayer formation.

Ag/AgCl electrodes with 1.5% agarose/2 M KCl bridges were used to apply the transmembrane voltage (V) and measure the current flowing through the bilayer (I). “Negative voltage” means that the side of protein addition is negative with respect to the other side. The transmembrane current measurements were carried out using an Axopatch 200B amplifier (Molecular Devices, LLC, Orleans Drive, Sunnyvale, CA, USA) in the voltage-clamp mode. Data were digitized by a Digidata 1550A and analyzed using pClamp 10.7 (Molecular Devices, LLC, Orleans Drive, Sunnyvale, CA, USA) and Origin 7.0 (OriginLab Corporation, Northampton, MA, USA). Data were acquired at a sampling frequency of 5 kHz using low-pass filtering at 1 kHz. Recordings of typical time courses of the transmembrane current flowing through the bilayer in the presence of the R44K^S^ peptide were filtered by a low-pass 8-pole Bessel filter (Model 9002, Frequency Devices, Ottawa, IL, USA) at 100 kHz.

### 4.5. Determination of Toxicity of Peptides against Human Fibroblasts and BT474

The peptides were pre-incubated for 12 h in DMEM medium without serum in a CO_2_ incubator. After the incubation period, serum was added to the medium to a final concentration of 10%, and the preparation was added dropwise onto the cells. The cytotoxic effects of peptides against eukaryotic cells (human fibroblasts and BT474) were subsequently assessed using the same method as in the previous study [62].

### 4.6. Statistical Analysis

Statistical analysis was conducted using the SigmaPlot v15 software package (SPSS Inc., Chicago, IL, USA). Each experiment was repeated at least twice (*n* ≥ 2). Data are presented as mean  ±  standard deviation (M ± SD). Statistical significance was assessed using Student’s *t*-test and analysis of variance (ANOVA) in the tests of the antibacterial activity of the peptides. The confidence interval value of 0.95 was used, corresponding to an alpha value of 0.05. Statistically significant decreases in optical density compared to the negative control were considered for specific peptide concentrations with *p*-values below 0.05.

### 4.7. In Silico Modeling

For the prediction of our structures, we used AlphaFold Colab [50] for multimer prediction. Colab uses the AlphaFold model parameters which are subject to the Creative Commons Attribution 4.0 International (CC BY 4.0 https://creativecommons.org/licenses/by/4.0/, accessed on 5 May 2024) license. Colab itself is provided under the Apache 2.0 license. The multiple sequence alignment (MSA) technique was used, where we see how well each residue is covered by similar sequences in the MSA.

The run_relax function was used, which allows for some stereochemical deviations. To obtain a more reliable model, multimer_model_max_num_recycles was set to 20. For the visualization of 3D structures, we used the UCSF Chimera 1.11.1 program.

## 5. Conclusions

In our study, peptides R23F^S^*, V31K^S^, R44K^S^, V31K^S^*, and R44K^S^* exhibited a strain-specific antimicrobial effect against *S. aureus*, MRSA, *P. aeruginosa*, and *E. coli*. Overall, these peptides inhibited the growth of bacterial strains 209P (*S. aureus*), ATCC 43300 (MRSA), ATCC 28753 (*P. aeruginosa*), and K12 (*E. coli*) at lower concentrations compared to those required to exhibit antimicrobial effects against strains 129B, SA 180, 2943, and MG1655 of the corresponding bacterial species. The amino acid modifications of peptides V31K^S^* and R44K^S^* resulted in a stronger suppression of the growth of the same strains of *S. aureus*, MRSA, *B. cereus*, *P. aeruginosa*, and *E. coli* compared to the unmodified peptides V31K^S^ and R44K^S^. In most cases, peptides R23F^S^*, R44K^S^, and R44K^S^*, containing the Tat penetration peptide fragment, inhibited bacterial growth at lower concentrations compared to peptides V31K^S^ and V31K^S^*, which only contained the Antp penetration peptide fragment. Peptide R44K^S^*, which inhibited the growth of most microorganisms used in this study, also exhibited the highest toxic effect on human fibroblasts and lung carcinoma cells compared to other peptides. Thus, based on the amyloidogenic region of the S1 protein, we developed peptides with improved antimicrobial properties by combining them with penetration peptide fragments and amino acid modifications.

## Figures and Tables

**Figure 1 ijms-25-06030-f001:**
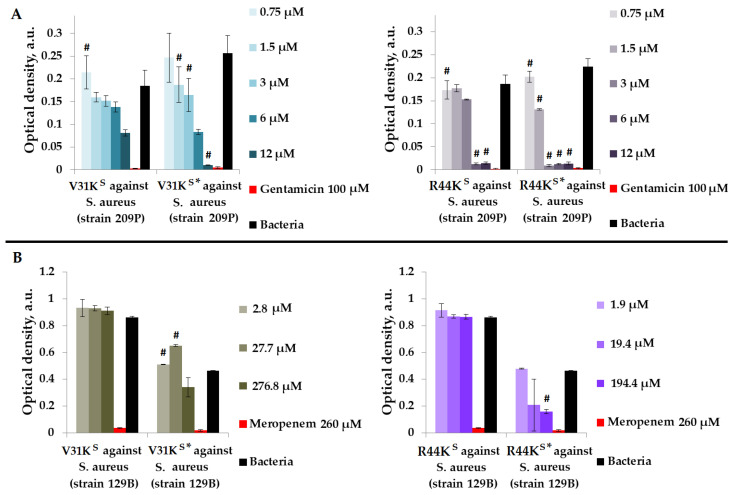
V31K^S^, R44K^S^, V31K^S^*, and R44K^S^* against *S. aureus* 209P (**A**) and 129B (**B**) strains after 15 h (**A**) and 24 h (**B**) of incubation. Bacterial cultures in a liquid medium served as negative controls. Gentamicin sulfate and meropenem antibiotics were used as positive controls. Error bars show standard errors. Number of independent experiments is two. #—for significant differences with negative control *p* < 0.05.

**Figure 2 ijms-25-06030-f002:**
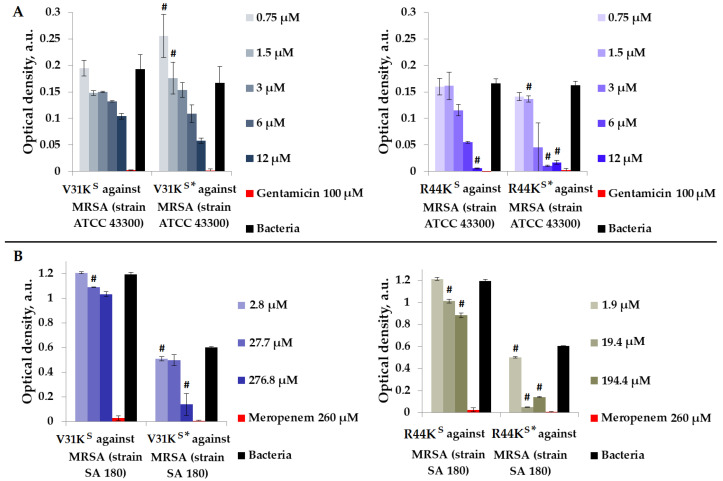
V31K^S^, R44K^S^, V31K^S^*, and R44K^S^* against MRSA ATCC 43300 (**A**) and SA 180 (**B**) strains after 15 h (**A**) and 24 h (**B**) of incubation. Bacterial cultures in a liquid medium served as negative controls. Gentamicin sulfate and meropenem antibiotics were used as positive controls. Error bars show standard errors. Number of independent experiments is two. #—for significant differences with negative control *p* < 0.05.

**Figure 3 ijms-25-06030-f003:**
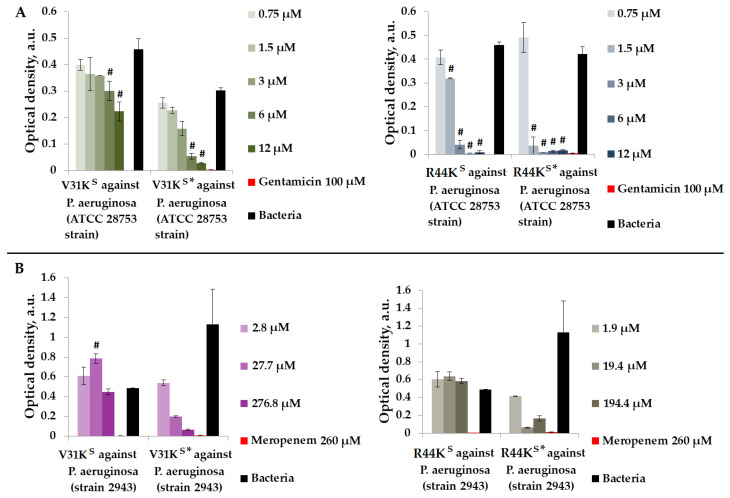
V31K^S^, R44K^S^, V31K^S^*, and R44K^S^* against *P. aeruginosa* ATCC 28753 (**A**) and 2943 (**B**) strains after 15 h (**A**) and 24 h (**B**) of incubation. Bacterial cultures in a liquid medium served as negative controls. Gentamicin sulfate and meropenem antibiotics were used as positive controls. Error bars show standard errors. Number of independent experiments is two. #—for significant differences with negative control *p* < 0.05.

**Figure 4 ijms-25-06030-f004:**
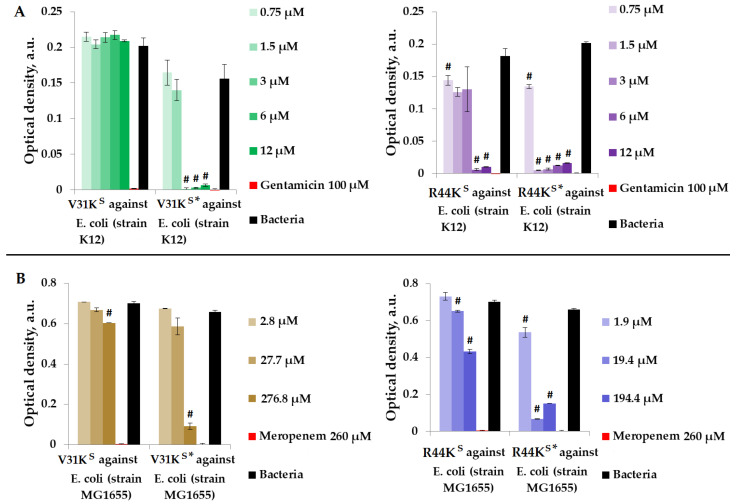
V31K^S^, R44K^S^, V31K^S^*, and R44K^S^* against *E. coli* K12 (**A**) and MG1655 (**B**) strains after 15 h (**A**) and 24 h (**B**) of incubation. Bacterial cultures in a liquid medium served as negative controls. Gentamicin sulfate and meropenem antibiotics were used as positive controls. Error bars show standard errors. Number of independent experiments is two. #—for significant differences with negative control *p* < 0.05.

**Figure 5 ijms-25-06030-f005:**
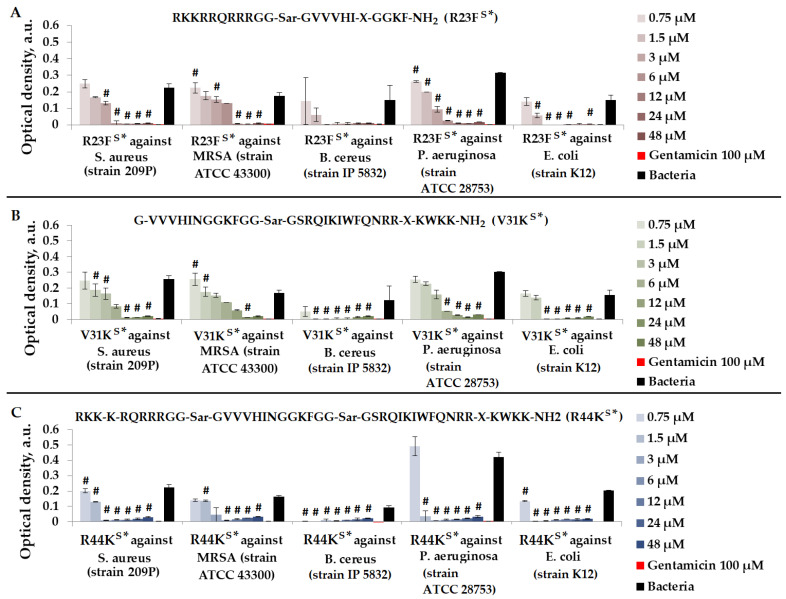
Results of R23F^S^* (**A**), V31K^S^* (**B**), and R44K^S^* (**C**) test for antimicrobial effect against *S. aureus* (strain 209P), MRSA (strain ATCC 43300), *B. cereus* (strain IP 5832), *P. aeruginosa* (strain ATCC 28753), and *E. coli* (strain K12) after incubation for 15 h. The antibiotic gentamicin sulfate was used as a positive control. The concentration of gentamicin was 100 µM. Cell cultures in a liquid medium (Mueller–Hinton Broth) served as a negative control. Error bars show standard errors. Number of independent experiments is two. #—for significant differences with negative control *p* < 0.05.

**Figure 6 ijms-25-06030-f006:**
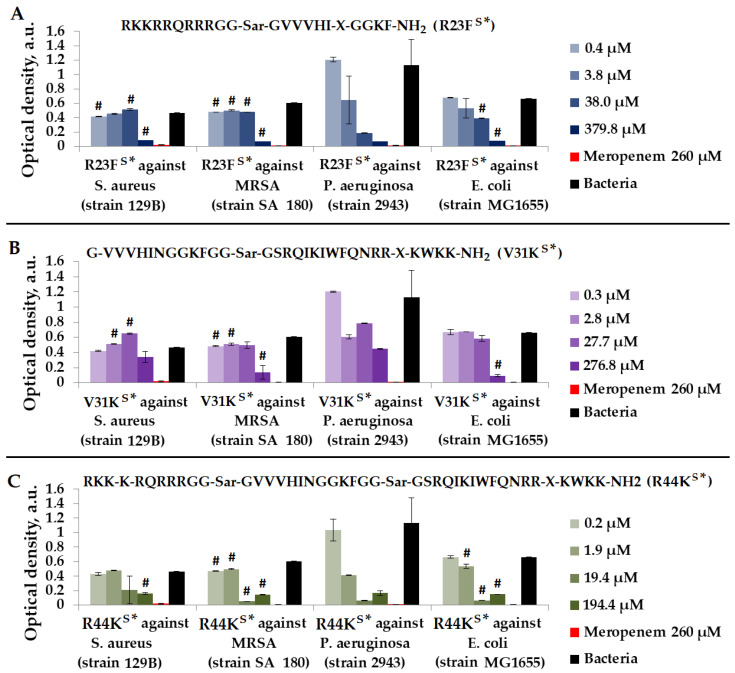
Results of R23F^S^* (**A**), V31K^S^* (**B**), and R44K^S^* (**C**) tests for antimicrobial effect against *S. aureus* (strain 129B), MRSA (strain SA 180), *P. aeruginosa* (strain 2943), and *E. coli* (strain MG1655) after incubation for 24 h. The antibiotic meropenem was used as a positive control. The concentration of meropenem was 260 µM. Cell cultures in a liquid medium (Mueller–Hinton Broth) served as a negative control. Error bars show standard errors. Number of independent experiments is two. #—for significant differences with negative control *p* < 0.05.

**Figure 7 ijms-25-06030-f007:**
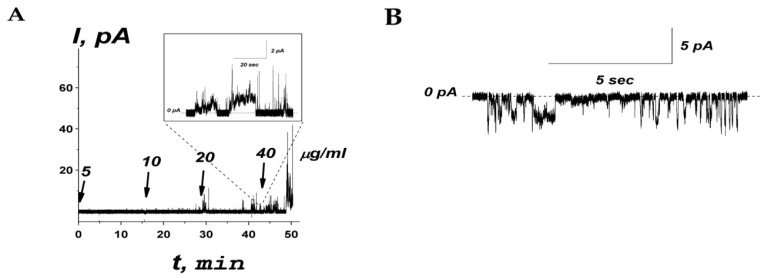
(**A**) An example of time course of the current flowing through the model membrane composed of POPE:POPG (50:50 mol%) produced by subsequent addition of R44K^S^ into bilayer bathing solution (0.1 M KCl, pH 7.4) up to 5, 10, 20, and 40 µg/mL (1.9, 3.7, 7.5, and 15.0 µM, respectively). The moments of peptide addition are indicated by the arrows. The corresponding concentrations of the peptide in the membrane-bathing solution are shown above the arrows. Inset: Single step-like transmembrane current fluctuations induced by addition of R44K^S^ at 20 µg/mL. The transmembrane voltage was equal to 100 mV. (**B**) An example of current fluctuations related to ion-permeable transmembrane pores produced by 20 µg/mL of R44K^S^ at −150 mV. The lipid bilayer was composed of POPE:POPG and bathed in 0.1 M KCl, pH 7.4. The dotted line corresponds to zero current.

**Figure 8 ijms-25-06030-f008:**
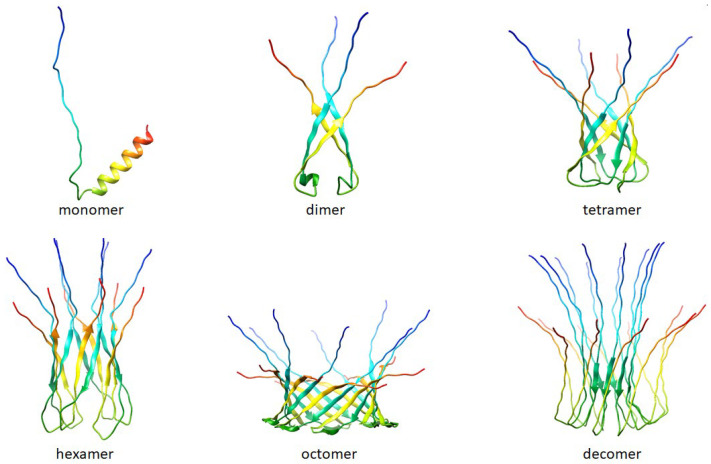
Folding patterns of peptide R44K^S^ were predicted by AlphaFold 2 [50] (AlphaFold 2. Available online: https://github.com/deepmind/alphafold (accessed on 18 April 2024)). The predictions are made by the “Colab” version of AlphaFold2; this notebook does not use templates (homologous structures) but uses a selected part of the Big Fantastic Database (BFD).

**Figure 9 ijms-25-06030-f009:**
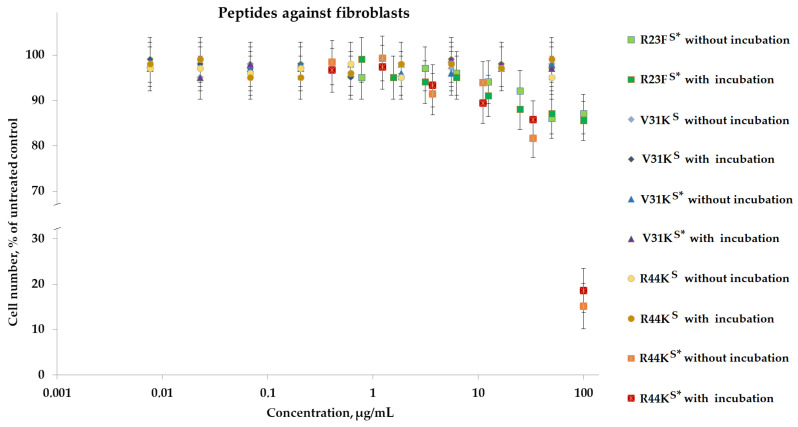
Effects of peptide treatment on survival of human fibroblasts. Error bars show the standard deviation, and the number of independent experiments is two.

**Figure 10 ijms-25-06030-f010:**
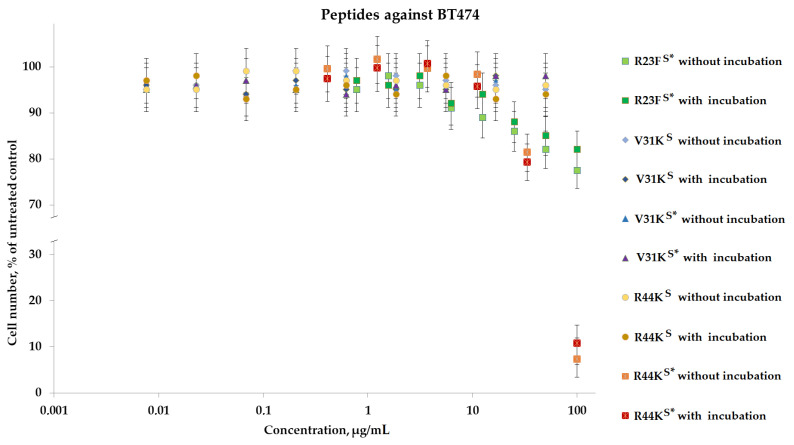
Effects of peptide treatment on survival of human breast tumor cell line BT-474. Error bars show the standard deviation, and the number of independent experiments is two.

**Table 1 ijms-25-06030-t001:** Results of testing peptides against diverse strains of pathogenic microorganisms.

Tested Peptide	Microorganisms	Peptide Efficacy in Liquid Medium after 15 h of Incubation, µM, against Bacterial Strains	Peptide Efficacy in Liquid Medium after 24 h of Incubation, µM, against Bacterial Strains
R23F^S^*	*S. aureus*	6 µM, 209P strain	379.8 µM, 129B strain
	MRSA	12 µM, ATCC 43300 strain	379.8 µM, SA 180-F strain
	*B. cereus*	3 µM, IP 5832 strain	No data
	*P. aeruginosa*	6 µM, ATCC 28753	38 µM, 2943 strain
	*E. coli*	3 µM, K12 strain	379.8 µM, MG1655 strain
V31K^S^	*S. aureus*	>12 µM, 209P strain	>276.8 µM, 129B strain
	MRSA	>12 µM, ATCC 43300 strain	>276.8 µM, SA 180-F strain
	*B. cereus*	No data	No data
	*P. aeruginosa*	>12 µM, ATCC 28753	>276.8 µM, 2943 strain
	*E. coli*	>12 µM, K12 strain	>276.8 µM, MG1655 strain
V31K^S^*	*S. aureus*	12 µM, 209P strain	>276.8 µM, 129B strain
	MRSA	24 µM, ATCC 43300 strain	276.8 µM, SA 180-F strain
	*B. cereus*	1.5 µM, IP 5832 strain	No data
	*P. aeruginosa*	12 µM, ATCC 28753	>276.8 µM, 2943 strain
	*E. coli*	3 µM, K12 strain	276.8 µM, MG1655 strain
R44K^S^	*S. aureus*	6 µM, 209P strain	>194.4 µM, 129B strain
	MRSA	12 µM, ATCC 43300 strain	>194.4 µM, SA 180-F strain
	*B. cereus*	No data	No data
	*P. aeruginosa*	3 µM, ATCC 28753	>194.4 µM, 2943 strain
	*E. coli*	6 µM, K12 strain	194.4 µM, MG1655 strain
R44K^S^*	*S. aureus*	3 µM, 209P strain	194.4 µM, 129B strain
	MRSA	6 µM, ATCC 43300 strain	19.4 µM, SA 180-F strain
	*B. cereus*	0.75 µM, IP 5832 strain	No data
	*P. aeruginosa*	1.5 µM, ATCC 28753	19.4 µM, 2943 strain
	*E. coli*	1.5 µM, K12 strain	19.4 µM, MG1655 strain

## Data Availability

Data contained within the article.

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
