# Peer review of "Optimizing Antimicrobial Peptide Design: Integration of Cell-Penetrating Peptides, Amyloidogenic Fragments, and Amino Acid Residue Modifications"

_ijms, 2024, doi:10.3390/ijms25116030_

Round 1

Reviewer 1 Report

Comments and Suggestions for Authors

Abstract

Line 35# Why compare gentamicin and meropenem? As the antimicrobial efficiency of the peptides is higher

Line 36# ranging from 24-48 μM how do the authors assess this range for the test?

Line 36# strongest is not a suitable word to use

Line 37# displayed superior efficacy

Keywords

Toxicity is not suitable

Introduction

Line 47 to 49# rephrase this paragraph

Line 56# is the synthetic AMPs compatible with the cell structure explaining the compatibility and suitability with the host cell.

Comments line 57# as the synthetic AMPs, how the pathogens modify themselves to it. Please explain.

Comments: combine the synthetic AMP literature into the same paragraphs

Comments: the synthetic AMPs more effective than direct drug delivery? Which one is more promising to explain in the advantages

Line 107 to 108# remove the sequence

Results

Better use the statistical tools to analyze data.

Remove the trend lines from the figures

Are the peptides lipophilic?

Discussion

Line 311# rearrange this

Major concern

No validation test is run for the isolation justification of peptides including Western blots. It will make the study more effective.

I suggest using moderate language to explain the outcomes.

Comments on the Quality of English Language

Minor editing of English language required

Reviewer 2 Report

Comments and Suggestions for Authors

The manuscript describes the investigation of the antimicrobial efficacy of peptides R23FS*, V31KS*, and R44KS*, which were engineered to include an amyloidogenic fragment sourced from the S1 protein of S. aureus, along with cell-penetrating peptide (CPP) and amino acid residue modifications.

The topic is of practical significance, the scope is adequate, and methods are appropriate.

However, some details in methods and results could be beneficial to increased reproducibility and improved scientific rigor, especially in the structural identity of the modified peptides, as per claimed functional modifications, providing molecular structural evidence that the targeted structural modification was achieved. The manuscript in the current state lacks such evidence. Effort towards this provided by Fig. 10 presents more confusion due to lack of experimental details of in silico modeling, and if this is a result it should be presented in Section 2 rather than 3.

Section 4.6 needs more information, such as confidence interval and alpha value.

The title is long and effusive, please condense.

All plots should be redone to change y-axis range to eliminate the large blank space. Fig. 8, 9 data points are hard to see, change y range, use break in axis.

Overall, there are too many data points and it is hard to extract the key information as described in Line 344-369. Suggest plot only the key concentrations that highlight the enhanced antimicrobial.

The authors make mention of cell-penetrating and anti-microbial properties and thus should explicit mention the specific role of glutamine peptides, which even on its own shows structure-specific properties supporting the objectives of the work under review.

Perhaps best is to mention this up-front in the introduction:

"Conjugating AMPs with cell-penetrating peptides (CPPs) [25] or antibiotics [26] enhances antibacterial activity, with L-glutamine in particular demonstrating the ability to raise efficacy of relevant anti-microbials[REF-1], arising from its fundamental structure and properties[REF-2]."(Page 2, Introduction, 2nd Paragraph, 1st sentence)

New-References:

REF-1: Nonhemolytic Cell-Penetrating Peptides: Site Specific Introduction of Glutamine and Lysine Residues into the α-Helical Peptide Causes Deletion of Its Direct Membrane Disrupting Ability but Retention of Its Cell Penetrating Ability

REF-2: An ab initio exploratory study of side chain conformations for selected backbone conformations of N-acetyl-l-glutamine-N-methylamide

Check Line 285-287, symbol “и”

Round 2

Reviewer 2 Report

Comments and Suggestions for Authors

Majority of the issues raised have been addressed, except for the lack of experimental details of in silico modeling.

PLease provide this in the Methods section to increase the reproducibility of the manuscript.

Author Response

Majority of the issues raised have been addressed, except for the lack of experimental details of in silico modeling.

PLease provide this in the Methods section to increase the reproducibility of the manuscript.

Answer: Thank you. We have added experimental details of in silico modeling in the Method section.